# TEXT TO IMAGE FOR MULTI-LABEL IMAGE RECOGNITION WITH JOINT PROMPT-ADAPTER LEARNING

## ABSTRACT

Benefited from image-text contrastive learning, pre-trained vision-language models, *e.g.,* CLIP, allow to directly leverage texts as images (TaI) for parameter-efficient fine-tuning (PEFT). While CLIP is capable of making image feature to be similar with the corresponding text features, modality gap remains a nontrivial issue and limits the MLR performance of TaI. Using multi-label image recognition (MLR) as an example, we present a novel method, called T2I-PAL to tackle the modality gap issue when using only text captions for PEFT. The core design of T2I-PAL is to leverage pretrained text-to-image generation models to generate photo-realistic and diverse images from text captions, thereby being beneficial for reducing modality gap. For better PEFT, we further combine both prompt tuning and adapter learning for enhancing classification performance. Extensive experiments on multiple benchmarks, including MS-COCO, VOC2007, and NUS-WIDE, show that our T2I-PAL can boost recognition performance by 3.47% in average above the top-ranked state-of-the-art methods. *Our code and models will be made publicly available.*

## 1 INTRODUCTION

In the recent few years, tremendous progress has been made in large-scale visual-language (VL) pre-trained models (Alayrac et al., 2022), *e.g.,* CLIP (Radford et al., 2021). Their promising performance has empowered a new learning paradigm for adapting VL pretrained models to various downstream tasks, *i.e.,* learning adapters or prompts in a parameter-efficient manner (Sun et al., 2022). In this work, we focus on a specific downstream task, *i.e.*, multi-label image recognition, which requires identifying all semantic labels included in an image (Sun et al., 2022; Chen et al., 2019c; Wang et al., 2017; Chen et al., 2019a).

When adapting VL pretrained models to MLR, one straightforward method is to annotate full semantic label sets for several images (see Fig. 1 (a)). Nonetheless, exhaustive annotation of MLR gives rise to a much higher cost. Fortunately, after large-scale contrastive learning, VL pretrained models have exhibited promising ability in aligning images with the corresponding text caption. Thus, we resort to using text captions as an alternative to images, *i.e.,* TaI-DPT (Guo et al., 2022), for learning prompts (see Fig. 1 (b)). Contrary to image data, text captions are not only easy to obtain but also explicitly provide the class labels, making them very encouraging for MLR.

However, existing VL pretrained models remain limited in their ability to entirely eliminate the modality gap in the feature space (Gu et al., 2022; Nukrai et al., 2022). Albeit several approaches have been suggested to mitigate this issue (Gu et al., 2022; Nukrai et al., 2022), we present an alternative solution by considering the breakthrough achievements in text-to-image generative models (Nichol et al., 2021; Ramesh et al., 2022; Ruiz et al., 2022; Saharia et al., 2022). In (Gu et al., 2022; Nukrai et al., 2022), noise injection is employed to the textual feature to alleviate the modality gap. On contrary, using text-to-image generation models, one can directly synthesize high-quality and diverse images from text captions. Thus, instead of extracting textual features from text captions, extracting image features from synthesized images, is expected to offer a natural solution for MLR.

In this paper, with a set of text captions, we suggest leveraging images synthesized by a text-to-image generation model and joint prompt-adapter, termed T2I-PAL, for MLR, see Fig. 1 (c). T2I-PAL does not require any original training images, nor does it suffer from less performance degradation due to the modality gap caused by using only text captions. To this end, we first crawl captions from

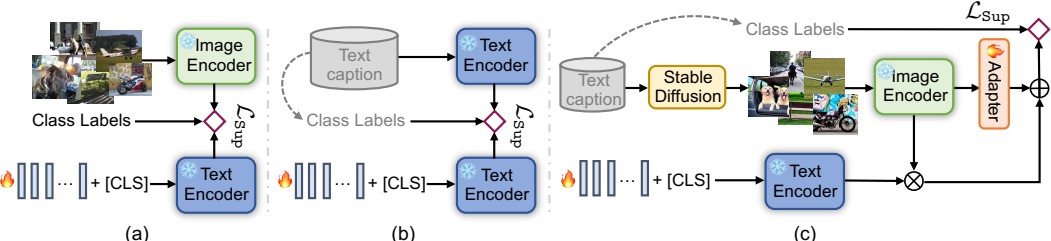

Figure 1: **Comparison** of **existing prompt tuning** methods and our T2I-PAL for MLR. **(a)** DualCoOp (Sun et al., 2022), which requires a substantial set of annotated images to learn the prompts, thereby being costly in annotation. **(b)** TaI-DPT (Guo et al., 2022), which solely leverages a set of text captions to learn the prompts, but suffers from the modality gap issue. In comparison, we present **(c)** T2I-PAL, where pretrained text-to-image generation model is used to tackle the modality gap, and joint prompt-adapter learning is adopted to improve MLR performance.

public datasets and filter textual descriptions containing one or more target object categories through a noun filter (Guo et al., 2022). Then, the text captions containing label information are fed into stable diffusion to obtain synthetic images. T2I-PAL replaces textual features with the image features from synthetic images without modifying the inherent mode of CLIP, thereby circumventing the modality gap of TaI-DPT (Guo et al., 2022). To enhance the classification performance, T2I-PAL combines both prompt tuning and adapter learning, and simultaneously absorbs the merits of TaI through a shared adapter between text and synthetic images. In particular, T2I-PAL achieves 6.3% improvement against TaI-DPT on the **MS-COCO** dataset. To sum up, our contributions are given as follows:

- We propose a novel prompt tuning method, termed T2-PAL, which aims to tackle the modality gap issue between text and image when using only text captions for effective MLR.

- T2I-PAL neither requires the original training images with full semantic annotation nor destroys the inherent mode of the CLIP model, allowing it to be embedded into any CLIP model.

- T2I-PAL combines both prompt tuning and adapter learning, and can absorb the benefits of TaI via a shared adapter between text and synthetic images.

- Experimental results show that T2I-PAL significantly outperforms the state-of-the-art MLR methods, and can be combined with existing prompting methods to further improve MLR performance.

## 2 RELATED WORK

**Multi-Label Image Recognition.** MLR aims to train a classifier that can recognize all object categories in an input image (Alfassy et al., 2019; Narayan et al., 2021; Simon et al., 2022). To establish the correlation between different labels, some works introduce graph neural networks (Chen et al., 2019c; Wang et al., 2020; Chen et al., 2019b; Zhao et al., 2021), recurrent neural networks (Wang et al., 2016a; 2017; Yazici et al., 2020), object proposals (Wang et al., 2016b; Liu et al., 2018), and attention mechanisms (Wang et al., 2017; Chen et al., 2018; 2019a) *etc.* into MLR to improve the accuracy of prediction. Another line of work proposes to solve MLR, where only partial labels are annotated per training image to reduce the annotation cost (Chen et al., 2019a;c; Durand et al., 2019; Chen et al., 2022; Pu et al., 2022). Inspired by the progress in large VL pre-trained models, DualCoOp leverages the strong alignment of textual and visual features pretrained by CLIP to learn positive and negative prompts with class names for MLR (Sun et al., 2022). Guo *et al.* alternated the text as images for prompt tuning, revising the default setting of learning prompts for images by visualizing data (Guo et al., 2022). Though impressive, the modality gap between the text and image makes it difficult for TaI to adapt to the image domain naturally in the test phase while learning prompts on the text (Guo et al., 2022). Instead of directly replacing the original image with text, we replace the text captions with synthetic images using text-to-image generation model. Therefore, we do not need to modify the input schema of the CLIP pre-trained model to make it perform better on downstream tasks.

**Parameter-Efficient Fine-tuning.** The parameter-efficient fine-tuning (PEFT) mechanism efficiently adapts the pretrained model to downstream tasks by updating only a small number of model parameters, thereby improving the efficiency of large models such as CLIP and reducing annotation and training costs (Zhu et al., 2022; Derakhshani et al., 2022; Wang et al., 2022; Mou et al., 2023; Sung

et al., 2022; Zhang et al., 2022). There are two mainstream methods: adapter, *e.g.,* T2I-Adapter (Mou et al., 2023), VL-Adapter (Sung et al., 2022), and Tip-Adapter (Zhang et al., 2022) and prompt, *e.g.,* CoOp (Zhou et al., 2022b), CoCoOp (Zhou et al., 2022a), ProDA (Lu et al., 2022), and TPT (Shu et al., 2022), which respectively refer to adding a small number of tokens and only updating a small number of new parameters in the model. For MLR, DualCoOp (Sun et al., 2022) and TaI-DPT (Guo et al., 2022) perform prompt tuning on the VL pre-training model in order to adapt the model to downstream tasks. However, in our method, due to the absence of fully annotated image training data, it is difficult to bridge the modality gap between text and images by only adding a small number of new parameters at the input to train multi-label classifiers on synthesized images. As a result, we suggest simultaneously introducing prompts and adapters in both input and model, where the adapter shares between the two modalities to explicitly enhance the classification performance on the synthesized image.

**Synthetic Data for Image Recognition.** The synthesized image has shown excellent performance in an increasing number of tasks due to its high flexibility (Choi et al., 2020; Rombach et al., 2022; Nichol et al., 2021; Sinha et al., 2021; Ho et al., 2022). Using early image generation methods, *e.g.,* VAEs (Kingma & Welling, 2013) and GANs (Goodfellow et al., 2020), initial attempts have been made in some visual tasks. In recent years, diffusion models have gradually become promising and powerful generative models that perform well in many applications, *e.g.,* high-resolution image synthesis (Rombach et al., 2022), text-to-image generation (Nichol et al., 2021; Ramesh et al., 2022), few-shot conditional image generation (Sinha et al., 2021), as well as point cloud generation (Ho et al., 2022). Several diffusion-based text-to-image models, including Stable Diffusion (Rombach et al., 2022), DALL-E2 (Ramesh et al., 2022), Imagen (Saharia et al., 2022), and GLIDE (Nichol et al., 2021), have been developed, providing an unprecedented synthesis quality and promoting the development of the AI-for-Art community. In particular, text-to-image generation can be seen as a conditional image generation task, which only requires inputting some natural language descriptions of what we want to express in the image and outputting it in visual. This motivates us to leverage images synthesized by text-to-image generation model for training MLR using only text captions.

## 3 METHOD

**Approach Overview.** In order to train a multi-label classifier by exploiting the large-scale pre-training models (CLIP (Radford et al., 2021)) in the absence of training images, we advocate leveraging synthesized images from text to image (T2I). With synthesized images, we do not need to modify the input modality of CLIP and thus can inherit the merits of large-scale visual language pre-training. Contrarily, treating text as image will reduce the performance of the CLIP model in downstream tasks due to the modality gap between text and image. As shown in Fig. 2 (a), T2I-PAL first constructs text captions from public image caption datasets (Lin et al., 2014; Krasin et al., 2017) as input to T2I for image synthesis (Guo et al., 2022). After that, we feed these synthesized images and the constructed text description into the image and text encoders, respectively, and freeze them. During training, we adopt three encoders from the pre-trained CLIP, *i.e.,* one image encoder to encode the synthesized image, two text encoders for the prompts and text captions. Additionally, to enhance the classification performance, T2I-PAL combines both prompt tuning and adapter learning, and can absorb the benefits of TaI via a shared adapter between text and synthetic images. During testing, the learned prompts are encoded by a text encoder to output the class embeddings, see Fig. 2 (b). The other text and visual encoders are replaced by a visual encoder and learned adapter, which receive test images as input and extract the features of each test image. These features are combined with global and local prompts to generate class embeddings through cosine similarity to give the final classification result.

**Preparation of Text Captions to Synthesize Images.** Following (Guo et al., 2022), we extract captions from public image captioning datasets (*e.g.,* MS-COCO (Lin et al., 2014)) and localized narratives from object detection datasets (*e.g.,* OpenImages (Krasin et al., 2017)) for generating the synthetic image. Among them, the labels used for training are also extracted from the caption, enabling no information from pictures to be disclosed during training (Guo et al., 2022). Concretely, given a multi-label dataset $\mathcal{X}$ with a set of object categories $\mathcal{C} = \{c_1, c_2, c_3, \ldots, c_C\}$, where $C$ is the number of categories, we first map nouns with similar meanings to their corresponding category labels through the noun filter, and then search for sentences containing at least one class name $c_i$ in $\mathcal{C}$, otherwise remove them directly. With the constructed text description, we feed it into a pre-trained

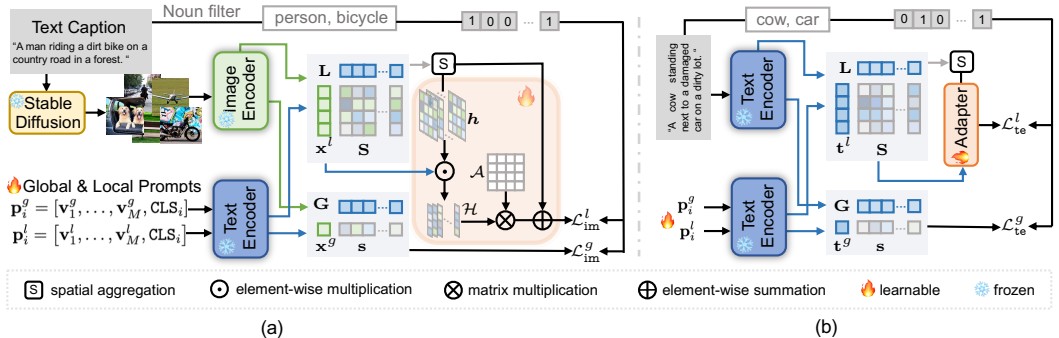

Figure 2: **Overall pipeline** of our proposed T2I-PAL. **(a)** Using pretrained text-to-image generation models (Rombach et al., 2022) to generate synthesized images from text captions and jointly learning prompt-adapter without modifying the inherent mode of the pertained CLIP. **(b)** Sharing adapter in the text caption branch to further enhance the classification performance.

T2I generation model, stable diffusion (Rombach et al., 2022), to synthesize its corresponding visual data $\mathcal{X}'$. In this way, we can use the paired labels and synthetic data to finetune the pre-trained model, *i.e.,* CLIP (Radford et al., 2021), in a parameter-efficient manner.

**T2I for Joint Prompt-Adapter Learning.** Following (Guo et al., 2022), to identify all labeled objects in the MLR task, global and local prompts are added to the class tokens to achieve discrimination by the coarsest-grained and fine-grained features, respectively. Formally, we have

$$
\begin{aligned}
\mathbf{p}_i^g &= [\mathbf{v}_1^g, \mathbf{v}_2^g, \mathbf{v}_3^g, \ldots, \mathbf{v}_M^g, \mathrm{CLS}_i], \\
\mathbf{p}_i^l &= [\mathbf{v}_1^l, \mathbf{v}_2^l, \mathbf{v}_3^l, \ldots, \mathbf{v}_M^l, \mathrm{CLS}_i],
\end{aligned}
\tag{1}
$$

where $\mathbf{p}^g$ and $\mathbf{p}^l$ are the global and local prompts consist with the learnable embedding $\mathbf{v}_j$, $\mathbf{v}_j'$, $j \in \{1, \ldots, M\}$ and class tokens $\mathrm{CLS}_i$ with $i$-th class in $C$ categories. As shown in Fig. 2, we then use the pretrained CLIP text encoder $\mathrm{E_{te}}$ to generate the global $\mathbf{G} = \{\mathbf{G}_i\}_{i=1}^C$ and local $\mathbf{L} = \{\mathbf{L}_i\}_{i=1}^C$ class embeddings. Accordingly, we extract global $\mathbf{x}^g \in \mathbb{R}^{N_{\mathrm{im}} \times D}$ and local $\mathbf{x}^l \in \mathbb{R}^{N_{\mathrm{te}} \times D}$ visual features from the pretrained CLIP visual encoder $\mathrm{E_{im}}$ with a visual input of synthetic image $\mathbf{x}$, where the $\mathbf{x}^l$ is the feature map before the attention pooling layer of CLIP. $N_{\mathrm{im}}$ and $N_{\mathrm{te}}$ are the image size and length of text tokens, respectively. Analogously, we copy a text encoder $\mathrm{E_{te}}$ of CLIP with a textual input of a piece of training text description to generate the global $\mathbf{t}^g \in \mathbb{R}^D$ and local $\mathbf{t}^l \in \mathbb{R}^D$ text features.

Given this, the global and local similarities of both visual and text features can be obtained by

$$
\mathbf{s}_i = \langle \boldsymbol{f}^g, \mathbf{G}_i \rangle, \quad \mathbf{S}_{ij} = \langle \boldsymbol{f}_j^l, \mathbf{L}_i \rangle,
\tag{2}
$$

where $\boldsymbol{f}^g$ and $\boldsymbol{f}^l$ indicate the global (*i.e.,* $\mathbf{x}^g$ or $\mathbf{t}^g$) and local (*i.e.,* $\mathbf{x}^l$ or $\mathbf{t}^l$) features from the pretrained CLIP encoder. We then aggregate the local similarities in a spatially weighted manner

$$
\mathbf{s}_i' = \sum_{j=1}^{N_*} \frac{\exp(\mathbf{S}_{ij}/\tau)}{\sum_{j=1}^{N_*} \exp(\mathbf{S}_{ij}/\tau)} \cdot \mathbf{S}_{ij},
\tag{3}
$$

where $N_*$ is the size of the image or length of text tokens, $\tau$ refers to the ability to focus on a specific location.

For enhancing classification performance, we further combine both prompt tuning and adapter learning. In particular, Zhang *et al.* proposed the Tip-Adapter (Zhang et al., 2022) by constructing query-key cache model from few-shot supervisions that can provide better vision-language modeling. With this perspective, we establish an adapter with query-key pairs, *i.e.,* local features treated as query, an initialized matrix with a size of $D \times C$ treated as key, that can be shared with both the visual and text features.

As shown in Fig. 2, the local text and visual features, *i.e.,* $\mathbf{t}^l$ and $\mathbf{x}^l$, that extracted by $\mathrm{E_{te}}$ and $\mathrm{E_{im}}$ serves as query for retrieving from the learanble matrix $\mathcal{A} \in \mathbb{R}^{C \times D}$. Note that the $\mathcal{A}$ is shared within the text and image branches. To enhance the network's ability to recognize multiple classes, we introduce a method that leverages a class-wise heat map to enrich the representation of local

visual features $\boldsymbol{f}_j^l$ across different classes. Given the local class embeding $\mathbf{L}_i$, we calculate the local similarity $\mathbf{S}_{ij}$ as their inner product. Then, class-wise heat map can be derived as

$$\boldsymbol{h}_{ij} = \frac{\exp\left(\mathbf{S}_{ij}/\tau\right)}{\sum_{j=1}^{N_*} \exp\left(\mathbf{S}_{ij}/\tau\right)}. \tag{4}$$

Class-wise heat map contributes to enhancing classification performance from two aspects: (1) Local similarities are fragile and noisy. Class-wise heat map can be used to aggregate the local similarities to obtain a robust class-wise similarity $\mathbf{s}_i' = \sum_{j=1}^{N*} \boldsymbol{h}_{ij} \cdot \mathbf{S}_{ij}$ (see Eq. 3). (2) Moreover, class-wise heat map can be used to obtain the class-wise attended feature $\mathcal{H}_i = \sum_{j=1}^{N_*} \boldsymbol{h}_{ij} \cdot \boldsymbol{f}_j^l$. Let $\mathcal{H} = [\mathcal{H}_1, \ldots, \mathcal{H}_i, \ldots, \mathcal{H}_C]$. We further introduce a learnable prototype matrix $\mathcal{A} = [\mathcal{A}_1, \ldots, \mathcal{A}_i, \ldots, \mathcal{A}_C]$, where $\mathcal{A}_i$ is the learnable prototype of class $i$. Consequently, we use

$$\boldsymbol{q} = \mathrm{diag}\left(\exp\left(-\beta\left(1 - \mathcal{H}\mathcal{A}^T\right)\right)\right), \tag{5}$$

to denote the affinity to the $i$-th class prototype, where $\beta$ refers the modulating hyper-parameter (Zhang et al., 2022). As noted, $\boldsymbol{f}^l$ can be either local CLIP text feature $\boldsymbol{t}^l$ or image feature $\boldsymbol{x}^l$. For $\boldsymbol{t}^l$ either or $\boldsymbol{x}^l$, we can use Eq. 5 to compute the class-wise affinity between $\mathcal{H}_i$ and $\mathcal{A}_i$. In terms of shared $\mathcal{A}$, we mean that the same $\mathcal{A}$ is adopted for both $\boldsymbol{t}^l$ and $\boldsymbol{x}^l$. By this way, both $\boldsymbol{t}^l$ and $\boldsymbol{x}^l$ contribute to better training of the learnable prototype matrix $\mathcal{A}$, thereby enhancing the class representation ability. Taking both $\mathbf{s}_i'$ and $\boldsymbol{q}_i$ into account, we define local predicted logits of the joint prompt-adapter as

$$\widetilde{\mathbf{s}}_i' = \alpha\boldsymbol{q}_i + \mathbf{s}_i', \tag{6}$$

where $\alpha$ is the residual ratio of the features of the CLIP's text or visual encoder. To sum up, class-wise heat map contributes to both local prompt (*i.e.*, $\mathbf{s}_i'$) and adapter (*i.e.*, $\boldsymbol{q}_i$). That is, T2I-PAL receives not only the prior knowledge of the pre-trained CLIP's visual encoder, but also the new knowledge that the adapter collects from the text. Accordingly, the smaller the value of $\alpha$, the more prior knowledge needs to be acquired from the pretrained CLIP's visual encoder, while the larger the value of $\alpha$, the more knowledge needs to be learned from the adapter. More importantly, as the learnable matrix $\mathcal{A}$ is shared with the synthetic image and text caption branches, it encourages the synthetic image to absorb the merits of the text caption, thereby enhancing the class representation ability of the classifier.

**Learning Objective.** The overall loss of our proposed T2I-PAL can be expressed as $\mathcal{L} = \gamma\mathcal{L}_{\mathrm{im}} + (1-\gamma)\mathcal{L}_{\mathrm{te}}$, where $\gamma$ is the trade-off weighted the two terms. Concretely, both the $\mathcal{L}_{\mathrm{im}}$ and $\mathcal{L}_{\mathrm{te}}$ contain two terms, *i.e.,* the global and local similarities in Eq. 2. Following (Guo et al., 2022), ranking loss (Gong et al., 2013) is adopted to measure the discrepancy between the classification score learned from a synthetic image and text caption with the ground-truth labels. The details can be formulated as

$$\begin{aligned} \mathcal{L}_*^g &= \sum_{i \in \{c^+\}} \sum_{j \in \{c^-\}} \max\left(0, \eta - \|\mathbf{s}_i - \mathbf{s}_j\|_2\right), \\ \mathcal{L}_*^l &= \sum_{i \in \{c^+\}} \sum_{j \in \{c^-\}} \max\left(0, \eta - \|\widetilde{\mathbf{s}}_i' - \widetilde{\mathbf{s}}_j'\|_2\right), \end{aligned} \tag{7}$$

where $\eta$ refers the margin value that determines the minimum amount by which the similarity score between positive classes should be greater than that of negative classes (Guo et al., 2022).

## 4 EXPERIMENTS

### 4.1 EXPERIMENTAL SETUP

**Implementation Details.** We implemented our method with Pytorch on one NVIDIA Tesla A100 GPU with 40GB of memory. The visual and text encoders are initialized from the CLIP pretrained model with ResNet-101 and Transformer, respectively. The impact of different visual encoders, *i.e.,* ResNet-50 and ResNet-101, on model performance also investigate in *Suppl*. During training, the pretrained encoder and decoder are frozen while only the prompts and adapters are optimized by the SGD (Kingma & Ba, 2014; Loshchilov & Hutter, 2017) with an epoch of 40 for all datasets. Concretely, the class-specific prompting is initialized with a Gaussian noise sampled from $\mathcal{N}(0, 0.02)$, where the length of both the global and local prompts is with a size of 16. We initialized the adapter with a size of $512 \times C$, where $C$ is the number of categories. The batch sizes are set to 64 and learning rates are initialized at 1e-4 for all three datasets. The hyperparameters of $\gamma$, $\alpha$, $\beta$, $\eta$, and $\tau_s$ are empirically set to 0.2, 1, 3.5, 1, and 0.02, respectively.

**Datasets.** Our proposed method T2I-PAL is evaluated on three datasets, *i.e.,* **VOC2007** (Everingham et al., 2010), **MS-COCO** (Lin et al., 2014), and **NUS-WIDE** (Chua et al., 2009). As there are no training images in our method, we adopt their official `test` set to evaluate our method. For `training`, we obtain the text captions from **MS-COCO** (Lin et al., 2014) for both **VOC2007** (Everingham et al., 2010), and **MS-COCO** (Lin et al., 2014) datasets. For **NUS-WIDE** (Chua et al., 2009), the localized narratives from OpenImages (Krasin et al., 2017) are adopted to cover all the concepts in this dataset.

**Baselines.** To investigate the effectiveness of our method, we conduct experiments on three scenarios, *i.e.,* `zero-shot` setting, `few-shot` setting, and `partial-label` setting. For the `zero-shot` setting, we compare our method with ZSCLIP (Radford et al., 2021), zero-shot CLIP model for MLR; and TaI (Guo et al., 2022), treating text as

Table 1: **Comparison** with state-of-the-arts under `zero-shot` setting, where ↑ indicates **improvements** compared with the **top-1** ranked baseline method, *i.e.,* TaI-DPT.

| Method | MS-COCO | VOC 2007 | NUS-WIDE |
|---|---|---|---|
| ZSCLIP[ICML21] | 47.3 | 76.2 | 36.4 |
| TaI-DPT[CVPR23] | 65.1 | 88.3 | 46.5 |
| T2I-PAL(Ours) | **71.4**(6.3) ↑ | **91.5**(3.2) ↑ | **47.4**(0.9) ↑ |

image in prompt tuning for MLR. For the `few-shot` setting, we compare our method with LaSO (Alfassy et al., 2019), ML-FSL (Simon et al., 2022); CoOp (Zhou et al., 2022b). For the `partial-label` setting, we compare our method with the following baselines: two graph-based methods, *i.e.,* SS-GRL (Chen et al., 2019a) and GCN-ML (Chen et al., 2019c); Par.BCE (Durand et al., 2019), SARB (Pu et al., 2022), and VL pre-trained model-based MLR methods, *i.e.,* DualCoOp (Sun et al., 2022) and +TaI-DPT (Guo et al., 2022).

Table 2: **Comparison** with state-of-the-arts under `partial-label` setting, where +TaI-DPT (Guo et al., 2022) and +T2I-PAL indicate integration with the MLR method, DualCoOp (Sun et al., 2022), respectively. ↑ indicates **improvements** compared with the **top-1** ranked baseline method, *i.e.,* +TaI-DPT (Guo et al., 2022).

| | Method | 10% | 20% | 30% | 40% | 50% | 60% | 70% | 80% | 90% | Avg. |
|---|---|---|---|---|---|---|---|---|---|---|---|
| **MS-COCO** | SSGRL[ICCV19] | 62.5 | 70.5 | 73.2 | 74.5 | 76.3 | 76.5 | 77.1 | 77.9 | 78.4 | 74.1 |
| | GCN-ML[CVPR19] | 63.8 | 70.9 | 72.8 | 74.0 | 76.7 | 77.1 | 77.3 | 78.3 | 78.6 | 74.4 |
| | Par.BCE[CVPR19] | 61.6 | 70.5 | 74.1 | 76.3 | 77.2 | 77.7 | 78.2 | 78.4 | 78.5 | 74.7 |
| | SARB[AAAI22] | 71.2 | 75.0 | 77.1 | 78.3 | 78.9 | 79.6 | 79.8 | 80.5 | 80.5 | 77.9 |
| | DualCoOp[NeurIPS22] | 78.7 | 80.9 | 81.7 | 82.0 | 82.5 | 82.7 | 82.8 | 83.0 | 83.1 | 81.9 |
| | +TaI-DPT[CVPR23] | 81.5 | 82.6 | 83.3 | 83.7 | 83.9 | 84.0 | 84.2 | 84.4 | 84.5 | 83.6 |
| | +T2I-PAL(Ours) | **82.7** | **83.0** | **84.5** | **84.6** | **84.8** | **85.0** | **85.6** | **85.8** | **85.9** | **84.7**(1.1) ↑ |
| **VOC 2007** | SSGRL[ICCV19] | 77.7 | 87.6 | 89.9 | 90.7 | 91.4 | 91.8 | 91.9 | 92.2 | 92.2 | 89.5 |
| | GCN-ML[CVPR19] | 74.5 | 87.4 | 89.7 | 90.7 | 91.0 | 91.3 | 91.5 | 91.8 | 92.0 | 88.9 |
| | Par.BCE[CVPR19] | 80.7 | 88.4 | 89.9 | 90.7 | 91.2 | 91.8 | 92.3 | 92.4 | 92.5 | 90.0 |
| | SARB[AAAI22] | 83.5 | 88.6 | 90.7 | 91.4 | 91.9 | 92.2 | 92.6 | 92.8 | 92.9 | 90.7 |
| | DualCoOp[NeurIPS22] | 90.3 | 92.2 | 92.8 | 93.3 | 93.6 | 93.9 | 94.0 | 94.1 | 94.2 | 93.2 |
| | +TaI-DPT[CVPR23] | 93.3 | 94.6 | 94.8 | 94.9 | 95.1 | 95.0 | 95.1 | 95.3 | 95.5 | 94.8 |
| | +T2I-PAL(Ours) | **93.7** | **94.8** | **94.8** | **94.9** | **94.9** | **95.2** | **95.5** | **95.5** | **95.5** | **95.0**(0.2) ↑ |
| **NUS** | DualCoOp[NeurIPS22] | 54.0 | 56.2 | 56.9 | 57.4 | 57.9 | 57.9 | 57.6 | 58.2 | 58.8 | 57.2 |
| | +TaI-DPT[CVPR23] | 56.4 | 57.9 | 57.8 | 58.1 | 58.5 | 58.8 | 58.6 | 59.1 | 59.4 | 58.3 |
| | +T2I-PAL(Ours) | **56.7** | **57.9** | **57.9** | **58.3** | **58.7** | **59.2** | **59.3** | **59.3** | **59.3** | **58.5**(0.2) ↑ |

## 4.2 COMPARISON WITH STATE-OF-THE-ARTS

Table 1 and Table 2 summarize the mAP values of the `zero-shot` and `partial label` settings over three datasets, *i.e.,* **VOC2007** (Everingham et al., 2010), **MS-COCO** (Lin et al., 2014), and **NUS-WIDE** (Chua et al., 2009), where +TaI-DPT indicates integrating TaI-DPT (Guo et al., 2022) with the partial-label MLR method, DualCoOp (Sun et al., 2022). As can be seen from these tables, the baseline performance of the partial labeled in Table 2 is generally better than the zero-shot baseline methods in Table 1, mainly because the labeled training data can make the model perform better on the test set. Under the `zero-shot` setting, the performance of our method on the three datasets is 6.3%, 3.2%, and 0.9% higher than the top-1 ranked method TaI-DPT (Guo et al., 2022), respectively. More importantly, even when integrated with the partial-label MLR method, DualCoOp (Sun et al., 2022), the performance of TaI-DPT (Guo et al., 2022) is still inferior to our method, *e.g.,* under 70% label annotation, 84.2 *vs.* **85.6** on **MS-COCO** (Lin et al., 2014), 95.1 *vs.* **95.5** on **VOC2007** (Everingham

Table 4: **Comparison** with state-of-the-arts under the `few-shot` setting, where +TaI-DPT (Guo et al., 2022) and +T2I-PAL indicate integration with the few-shot MLR method, CoOp (Zhou et al., 2022b), respectively. ↑ indicates **improvements** compared with the **top-1** ranked baseline method, *i.e.,* +TaI-DPT (Guo et al., 2022).

|  | Method | 0-Shot | 1-Shot | 2-Shot | 4-Shot | 8-Shot | 16-Shot |
|---|---|---|---|---|---|---|---|
| MS-COCO | ZSCLIP[ICML21] | 47.3 | – | – | – | – | – |
|  | CoOp[IJCV22] | – | 52.6 | 57.3 | 58.1 | 59.2 | 59.8 |
|  | TaI-DPT[CVPR23] | 65.1 | – | – | – | – | – |
|  | +TaI-DPT[CVPR23] | – | 65.8 | 66.2 | 67.6 | 68.1 | 68.9 |
|  | T2I-PAL(Ours) | **71.4**(6.3)↑ | – | – | – | – | – |
|  | +T2I-PAL(Ours) | – | **71.6**(5.8)↑ | **71.8**(5.6)↑ | **73.1**(5.5)↑ | **73.5**(5.4)↑ | **74.1**(5.2)↑ |
| VOC-2007 | ZSCLIP[ICML21] | 76.2 | – | – | – | – | – |
|  | CoOp[IJCV22] | – | 79.3 | 83.2 | 83.8 | 84.5 | 85.7 |
|  | TaI-DPT[CVPR23] | 88.3 | – | – | – | – | – |
|  | +TaI-DPT[CVPR23] | – | 88.6 | 89.2 | 89.1 | 89.5 | 90.1 |
|  | T2I-PAL(Ours) | **91.5**(3.2)↑ | – | – | – | – | – |
|  | +T2I-PAL(Ours) | – | **91.7**(3.1)↑ | **92.1**(2.9)↑ | **92.2**(3.1)↑ | **92.3**(2.8)↑ | **92.9**(2.8)↑ |

et al., 2010), and 58.6 *vs.* **59.3** on **NUS-WIDE** (Chua et al., 2009). These results support our conclusion that leveraging pretrained text-to-image generation models to generate photo-realistic and diverse images from text captions is beneficial for reducing the modality gap. We further evaluate the effectiveness of our T2I-PAL with the various state-of-the-arts in the `few-shot` setting. Following existing few-shot MLR methods (Alfassy et al., 2019; Simon et al., 2022), a model was trained on known classes while deployed to 16 novel classes. In Table 3, we record various few-shot MLR methods with zero-shot TaI-DP (Guo et al., 2022) and T2I-PAL on the 16 novel classes. As can be seen from this table, the performance of T2I-PAL is higher than that of TaI-DPT, even **surpassing** the top-1 rank method trained on 5 shot samples, *i.e.,* ML-FSL/5-shot (Simon et al., 2022): 63.6 *vs.* T2I-PAL/0-shot: **66.3**.

Additionally, following (Guo et al., 2022), we adopt the strategy in (Alfassy et al., 2019) to treat all classes as novel classes and select 1, 2, 4, 8, and 16 shot samples for each class. Since neither TaI-DPT (Guo et al., 2022) nor our method, T2I-PAL, has original annotated images, we integrate them with CoOp to record the performance under different `few-shot` settings, termed +TaI-DPT (Guo et al., 2022) and

Table 3: **Comparison** against various SOTAs under `few-shot` setting on **MS-COCO** dataset with 16 **novel** classes.

| Method | 0-Shot | 1-Shot | 5-Shot |
|---|---|---|---|
| LaSO[CVPR19] | – | 45.3 | 58.1 |
| ML-FSL[WACV22] | – | 54.4 | 63.6 |
| TaI-DPT[CVPR23] | 59.2 | – | – |
| T2I-PAL(Ours) | **66.3**(4.1)↑ | – | – |

+T2I-PAL. As can be seen in Table 4, although the integration of TaI-DPT (Guo et al., 2022) and CoOp (Zhou et al., 2022b), +TaI-DPT (Guo et al., 2022), improves the average performance of CoOp (Zhou et al., 2022b) in the few-shot of the two datasets, *i.e.,* $57.4 \rightarrow 67.3$ on **MS-COCO** (Lin et al., 2014), and $83.3 \rightarrow 89.3$ on **VOC2007** (Everingham et al., 2010), our method still gets further improvement, *i.e.,* $67.3 \rightarrow 72.8$ on **MS-COCO** (Lin et al., 2014), and $89.3 \rightarrow 92.2$ on **VOC2007** (Everingham et al., 2010). More importantly, the performance of our method on zero-shot has surpassed the performance on 16-shot after the integration of TaI-DPT and CoOp (Zhou et al., 2022b), *e.g.,* 68.9 *vs.* **71.4** on **MS-COCO** (Lin et al., 2014), and 90.1 *vs.* **91.5** on **VOC2007** (Everingham et al., 2010). The outstanding performance of T2I-PAL in the few-shot setting again confirms our core idea that learning prompts and adapters using synthesized images through text captions can help tackle the modality gap issue when using only text captions for PEFT.

## 4.3 ABLATION STUDIES

**Effective of Training Data.** Here, we investigate how the different types of `training` data influence the results. As such, we construct some variations in Table 5, where `Syn.Img`, `Text`, and `Ori.Img` indicate that the method uses the synthesized image, text caption, and original image as the `training` data, respectively. As can be seen from this table, the performance on the `Syn.Img` is significantly better than `Text`, *i.e.,* TaI-DPT: 88.3 *vs.* Ours: (`Syn.Img`): **90.8**. When our method employs two types of `training` data, the performance will be further improved, *i.e.,* Ours (`Syn.Img`): $90.8 \rightarrow$ Ours (Full): **91.5**. Additionally, Ours: (`Three`) can be as the upper bound (UB.) of our method which leverages text captions and synthetic images as well as original annotated images in training.

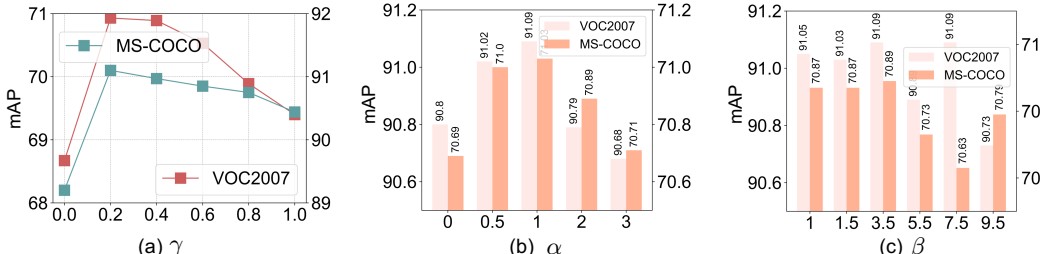

Figure 3: **Analysis** with regard to the different values of $\gamma$, $\alpha$, and $\beta$ on the two datasets, *i.e.,* **VOC 2007** and **MS-COCO**, where **(a)** the *larger* the value of the $\gamma$, the larger the proportion of the *synthetic image* in our method, vice versa; **(b)** the *smaller* the value of $\alpha$, the more *prior knowledge* needs to be acquired from the pretrained CLIP's visual encoder, vice versa; and **(c)** $\beta$ controls the *sharpness* of the affinities.

Nonetheless, the numbers of synthetic images is $6\times$ of the original images. Thus, further including original images in training (*i.e.,* `Ours: (Three)`, $91.6$) only brings moderate gain to `Ours (Full)` ($91.5$). The results indicate that original images adds minor diversity when the number of synthetic images are larger (*i.e.,* $6\times$).

**Analysis of Ratio $\gamma$.** Since our method absorbs the complementary advantages of text captions and synthesized images from the pretrained text-to-image model, it is necessary to investigate how these two mechanisms can help train classifiers. To this end, we performed an analysis of the effect of the hyperparameter $\gamma$ on the two datasets. The results are shown in Fig. 3 (a), where the

Table 5: **Ablation studies** with regard to the different types of `training` data, where (`UB.`) is the *upper-bound* performance by using text captions, synthetic images and original images.

| Variation | Syn.Img | Text | Ori.Img | **VOC 2007** |
|---|---|---|---|---|
| TaI-DPT | – | ✓ | – | $88.3_{(3.3)}\downarrow$ |
| DualCoOp | – | – | ✓ | $91.6$ |
| Ours (Syn.Img) | ✓ | – | – | $90.8_{(0.8)}\downarrow$ |
| Ours (Full) | ✓ | ✓ | – | $91.5_{(0.1)}\downarrow$ |
| Ours (Three)(UB.) | ✓ | ✓ | ✓ | $91.6$ |

larger the value of the $\gamma$, the larger the proportion of the synthesized image, and vice versa. We can see from the figure that when $\gamma = 0$, that is, there is no synthesized image component in our model, the performance is lowest. Such results indicate that the synthetic image is very important for PEFT without the original annotated training image. Training prompts on text captions for PEFT will be greatly affected by the modality gap because training prompts in the text domain are difficult to directly adapt to the image domain. However, the performance of `T2I-PAL` increases rapidly with increasing $\gamma$, which is mainly the gain brought by synthetic images, *i.e.,* directly making up the modality gap when using only text captions for PEFT. For example, when $\gamma = 0.2$, our method achieves a performance of $91.1$ and $70.9$ on the two datasets. When the $\gamma$ values continue to increase, *e.g.,* $\gamma = 1$, the performance of `T2I-PAL` decreases to $90.4$ and $69.4$, mainly because the advantages of text caption are lost while completely adopting the synthetic images.

Furthermore, we observe that different values of $\gamma$ perform similarly on the two datasets, which proves that `T2I-PAL` are robust to different datasets. To this end, we encourage adopting text captions and also synthetic images to tackle the modality gap under such a scenario. Further, we also explore the effect with additional text data as well as the quality of text captions in our method. Please refer to the *Suppl.* for details.

**Effect of PEFT.** As mentioned in the Sec. 3, we share the adapter on the local features between the text caption and synthetic image branches to enhance the class representation of the model. Therefore, here we examine how the adapter affects the effectiveness of our method from three aspects, *i.e.,* adapter on the global features, without adapter, and the hyperparameter analysis of the adapter. We first analyze whether using an adapter on global features can improve the performance of `T2I-PAL`. To this end, we constructed two variants of `T2I-PAL`, *w.* `Glo.Adp`, indicating that our method uses an adapter on global features, and *w/o.* `Adp`, indicating that our method does not contain the adapter module. Table 4 records the performance of these variants on the three datasets. As can be seen from the table that *w/o.* `Adp` has the worst performance on the three datasets. This suggests that sharing an adapter between two modalities is helpful for enhancing MLR. Additionally, although the performance of *w.* `Glo.Adp` has slightly improved, its performance is still far below that of our full model, `T2I-PAL`, *e.g., w.* `Glo.Adp`: $70.7$ *vs.* `Ours`: **71.4** on **MS-COCO**. This is attributed to the fact that more category knowledge can be captured on local features, thereby enhancing MLR. To this

Table 6: **Ablation studies** with regard to the different `Adapters` of `T2I-PAL` on the three datasets, where ↓ indicates **decrements** compared with our full model, *w.* `Loc.Adp` (`Ours`).

| Variation | Glo.Adp | Loc.Adp | MS-COCO | VOC-2007 | NUS-WIDE | Average |
|---|---|---|---|---|---|---|
| *w/o.* Adp | – | – | 70.6$_{(0.8)}$↓ | 91.0$_{(0.5)}$↓ | 47.1$_{(0.3)}$↓ | 69.57$_{(0.53)}$↓ |
| *w.* Glo.Adp | ✓ | ✓ | 70.7$_{(0.7)}$↓ | 91.1$_{(0.4)}$↓ | 47.1$_{(0.3)}$↓ | 69.63$_{(0.47)}$↓ |
| *w.* Loc.Adp (Ours) | – | ✓ | 71.4 | 91.5 | 47.4 | 70.1 |

end, our full model, *w.* `Loc.Adp` (`Ours`), shares an adapter module between local features of the two modalities, which can further tackle the modality gap issue when using only text captions for PEFT.

Here, we investigate two important hyperparameters closely related to the adapter in Eq. 5 and 6, *i.e.,* $\alpha$, the residual ratio of the features of the CLIP's text or visual encoder; and $\beta$, the modulating hyper-parameter that controls the sharpness of the affinities. We plot the histogram of different values of $\alpha$ in Fig. 3 (b), where the smaller the value of $\alpha$, the more prior knowledge needs to be acquired from the pretrained CLIP's visual encoder, while the larger the value of $\alpha$, the more knowledge needs to be learned from the adapter. As can be seen, when $\alpha = 0$, the model performs poorly because it degenerates into the zero-shot CLIP. As the value of $\alpha$ increases, the performance of the model starts to improve, with the highest results obtained at $\alpha = 1$, *i.e.,* $91.09$ and $71.03$.

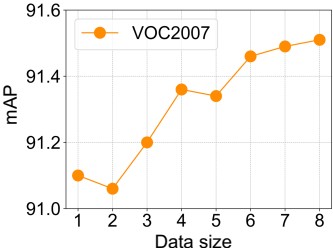

Figure 4: **Ablation studies** on the different **size of synthetic** data on our method.

However, when the value of $\alpha$ increases further, the performance of the model decreases, mainly because the prior knowledge from CLIP also plays an important role. That is, a good balance of knowledge learned from the adapter and prior knowledge that needs to be acquired from the pretrained CLIP's visual encoder can enable the model to achieve the highest performance. We then plot the histogram of different values of $\beta$ in Fig. 3 (c). If the value of $\beta$ is large, then the classification prediction for a test image is primarily affected by the training samples in its vicinity, and vice versa. As can be seen from Fig. 3 (c), when $\beta = 3.5$, our method achieves the highest performance. The results in Fig. 3 (a) and (b) also show that the performance of different $\alpha$ and $\beta$ values on the different datasets is consistent. Given this, we set $\alpha = 1$, $\beta = 3.5$ in our experiments.

**Effect of Synthetic Dataset Size**. Considering that synthetic data is freely available, photo-realistic, diverse, and not limited by the annotations, we examine whether more synthetic visual data can help improve model performance. To this end, we record the testing results on **VOC 2007** with synthetic visual data of different sizes in Fig. 4. It can be seen from the figure that as the number of synthetic images increases, the performance of our method, `T2I-PAL` gradually improves until it stabilizes when the size reaches 6. Given this, we set the size of the synthetic image to 6 in our experiments in Sec. 4.2. It is worth noting that even when the size of synthetic image equals 1, the performance of the model is still much higher than the top-1 ranked baseline method, TaI-DPT, *i.e.,* TaI-DPT (Guo et al., 2022): 88.3 *vs.* `Ours`: **91.1**, gains 3.2 improvements on **VOC 2007**. Consequently, the result indicates that our method provides an effective solution when using only text captions for MLR. Additionally, we explore the impact of the quality of the synthesized images and the modality gap between the two images on our method, see ***Suppl.*** for details.

## 5 CONCLUSION

This paper presented a new PEFT method, `T2I-PAL`, based on a large-scale pre-trained vision-language model to address the modality gap issue when performing PEFT only with text captions. The core design of `T2I-PAL` is to utilize a pre-trained text-to-image generation model to synthesize photo-realistic and diverse images from text captions. `T2I-PAL` provides two appealing benefits: 1) it does not require any full semantically annotated training image, thereby lowering the burden of manual annotation; 2) it does not destroy the inherent mode of the CLIP model and can be implanted into any CLIP model. Additionally, `T2I-PAL` combines both prompt tuning and adapter learning with the two modalities, thereby enhancing classification performance. Potential limitations of this work is that our paper relies on pretrained text-to-image generation models to generate photo-realistic and diverse images from text captions. However, it does not address potential limitations or challenges associated with these models, such as biases in the generated images or their capability to faithfully represent the intended visual content.

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
