## A   APPENDIX

## CONTENTS

The following items are included in our supplementary material:

## B   EFFECT WITH ADDITIONAL TEXT DATA.

We use Llama (Touvron et al., 2023) to generate extra text captions containing one or several category names. More details on leveraging Llama (Touvron et al., 2023) to generate text captions are provided in Fig. 5. Then, we use Stable Diffusion to generate the corresponding synthetic images. By this way, the size of text captions is doubled (*i.e.,* T2I-PAL w/ 2× text). Using **VOC 2007**, we provide the results of T2I-PAL and T2I-PAL w/ 2× text in Table 7, 8 and 9. Benefiting from pre-trained language models, the generated text captions can be of both high quality and high diversity. One can see that, the introduction of extra text captions consistently improves classification performance under zero-shot, partial-label and few-shot settings.

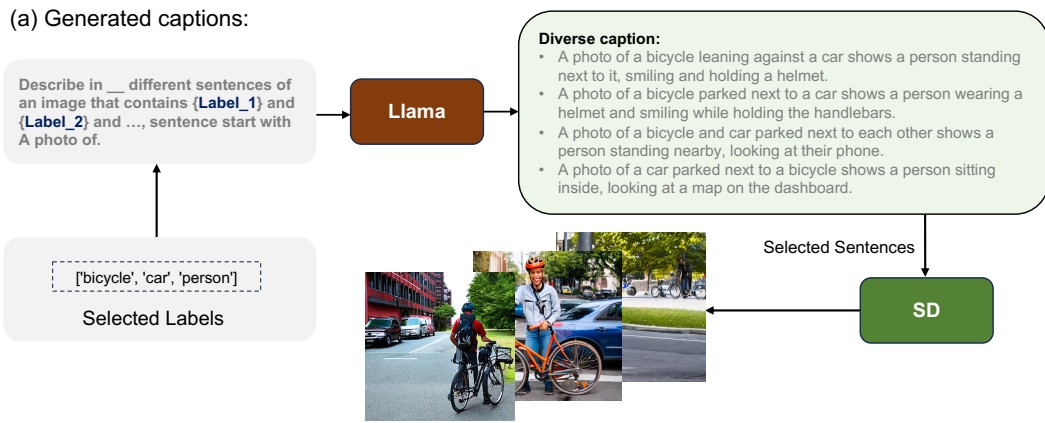

Figure 5: (a) Illustration of leveraging Llama to generate text captions; (b) Examples of VOC captions with the selected labels. As a comparison, Llama can also provide quite complex and diverse text captions.

Table 7: **Ablation studies** of zero-shot setting with additional text data on **VOC 2007**.

| Dataset | T2I-PALw/2× | T2I-PAL |
|---|---|---|
| **VOC 2007** | 91.8 | 91.5 |

Table 8: **Ablation studies** of `partial label setting` with additional text data on **VOC 2007**.

| | Method | 10% | 20% | 30% | 40% | 50% | 60% | 70% | 80% | 90% | **Avg.** |
|---|---|---|---|---|---|---|---|---|---|---|---|
| VOC | +T2I-PAL w/2× | 93.9 | 94.8 | 94.9 | 95.0 | 94.9 | 95.5 | 95.7 | 95.7 | 95.7 | 95.1 |
| | +T2I-PAL(Ours) | **93.7** | **94.8** | **94.8** | **94.9** | **94.9** | **95.2** | **95.5** | **95.5** | **95.5** | **95.0** |

Table 9: **Ablation studies** of `few-shot setting` with additional text data on **VOC 2007**.

| Method | **0-Shot** | **1-Shot** | **2-Shot** | **4-Shot** | **8-Shot** | **16-Shot** |
|---|---|---|---|---|---|---|
| +T2I-PAL w/2× | – | 91.9 | 92.2 | 92.3 | 92.5 | 93.0 |
| +T2I-PAL(Ours) | – | **91.7** | **92.1** | **92.2** | **92.3** | **92.9** |

## C  INFLUENCE OF THE QUALITY OF TEXT CAPTIONS.

For **COCO** and **VOC 2007**, we use the text captions provided in the datasets, while for **NUS** we acquire the necessary captions from the OpenImages. Thus, the quality of the constructed text descriptions can be ensured in our experiments. Besides, one can generate text captions using existing language models, *e.g.,* Llama (Touvron et al., 2023). In Fig. 5, we show the diversity of the generated text captions. In the Table 10, we also show the results using the text captions generated by Llama (Touvron et al., 2023). One can see that `Ours(Text.Llama)` achieves similar performance with `T2I-PAL`, which shows its robustness *w.r.t* the quality of the text description. Further, existing language models (*e.g.,* Llama) have been effective in generating text descriptions with satisfying quality. Thus, in practice, the quality of the caption is not a major concern for our method.

Table 10: **Ablation studies** of the quality of *text captions* influence our method on the three datasets.

| **Methods** | **COCO** | **VOC 2007** | **NUS** |
|---|---|---|---|
| Ours(Text.Llama) | 71.2 | 91.4 | 47.3 |
| Ours | 71.4 | 91.5 | 47.4 |

## D  INFLUENCE OF THE QUALITY OF SYNTHETIC IMAGES.

Since our method build on the SD, the quality of synthetic images also affects the model performance. In general, better classification performance may be attained when synthetic images are of higher quality. To illustrate this, we compare the performance using the synthetic images generated by SD2.0 (*i.e.,* `T2I-PAL`) and SD1.4 (*i.e.,* `Ours(SD1.4)`) in Table 11. The results show that better image quality gives rise to slightly better MLR performance.

## E  DISCUSSION ON THE MODALITY GAP BETWEEN ORIGINAL AND SYNTHETIC IMAGES

Albeit our `T2I-PAL` is effective, it is difficult to fully eliminate the modality gap. We provide the results of `Ours(Ori.Img)`, `Ours w/ 1× Syn.Img`, and `T2I-PAL` (*i.e.,* `Ours w/ 6× Syn.Img`) under zero-shot setting, across three datasets, in Table 12. As can be seen, `Ours w/ 1× Syn.Img` performs on par with `Ours(Ori.Img)`, indicating that SD is promising in generating photo-realistic images, and greatly minimize the modality gap. Furthermore, by using $6\times$ synthetic images, `T2I-PAL` outperforms `Ours(Ori.Img)` on all datasets, indicating that the diversity provided by more synthetic images compensates for the modality gap.

## F  EFFECT OF VISUAL ENCODERS.

Here, we investigate the impact of different visual encoders, *i.e.,* ResNet50 and ResNet101, on model performance. As such, we use ResNet50 and ResNet101 as visual encoders for TaI-DPT and T2I-PAL, respectively. The zero-shot results on the three datasets are summarized in Table 13. As listed in this table, there is almost no difference in the performance of TaI-DPT on the different

Table 11: **Ablation studies** of the quality of *synthetic images* influence our method on the three datasets.

| Methods | COCO | VOC 2007 | NUS |
|---------|------|----------|-----|
| Ours(SD1.4) | 71.2 | 91.2 | 47.2 |
| Ours | 71.4 | 91.5 | 47.4 |

Table 12: **Ablation studies** of elimination of the modality gap.

| Methods | COCO | VOC 2007 | NUS |
|---------|------|----------|-----|
| Ours(1×Syn.Img) | 70.5 | 90.8 | 46.8 |
| Ours(Ori.Img) | 70.7 | 90.9 | 46.9 |
| Ours(6×Syn.Img) | 71.4 | 91.5 | 47.4 |

visual encoders, *e.g.,* on the ResNet50, TaI-DPT Guo et al. (2022) obtain the performance of 88.3 on **VOC 2007**, 65.1 on **MS-COCO**, and 46.5 on **NUS-WIDE**; on the ResNet101, TaI-DPT Guo et al. (2022) obtains the performance of 88.3 on **VOC 2007**, 65.4 on **MS-COCO**, and 45.3 on **NUS-WIDE**. In particular, TaI-DPT Guo et al. (2022) decreases the performance on the **NUS-WIDE** from 46.5 to 45.3 when using ResNet101 as the visual encoder. On contrary, our method improves the performance on the three datasets when using ResNet101 as the visual encoder, *i.e.,* on the ResNet50, T2I-PAL obtains the performance of 88.8 on **VOC 2007**, 66.1 on **MS-COCO**, and 45.5 on **NUS-WIDE**; on the ResNet101, T2I-PAL obtains the performance of 91.5 on **VOC 2007**, 71.4 on **MS-COCO**, and 47.4 on **NUS-WIDE**. More importantly, even based on ResNet50, our method still achieves improvements over TaI-DPT Guo et al. (2022), *i.e.,* 0.5 gains on **VOC 2007** and 1.0 gains on **MS-COCO**. In summary, as TaI-DPT Guo et al. (2022) is based on text description, there is little difference in its performance for different visual encoders, while our method is based on synthetic images, making different visual encoders perform differently. Nevertheless, our method outperforms TaI-DPT Guo et al. (2022) on two datasets, *i.e.,* **MS-COCO** and **VOC 2007**, no matter if it is based on ResNet50 or ResNet101.

## G COMPLEMENTARITY OF TAI AND T2I.

As we have mentioned in Sec. 3 that our method absorbs the complementary merits of text captions and synthesized images from the pretrained text-to-image generation model, we list some examples in Fig. 6 to illustrate it intuitively. This figure lists the predictions of each class in each corresponding image, where T2I refers to our method but without text description, ✓ and ✗ refer to the **correct** and **wrong** predictions of each class, respectively. As can be seen from this figure, TaI-DPT Guo et al. (2022) produces wrong predictions in some categories, while T2I produced wrong predictions in the remaining categories, *e.g.,* in image **(a)**, TaI-DPT Guo et al. (2022) produces **wrong** predictions in the class of "person", while T2I produces **wrong** predictions in the class of "tvmonitor". Nonetheless, our proposed method, T2I-PAL, absorbs the complementary merits of text captions and synthesized images, thereby yielding correct predictions for these two categories, *i.e.,* "person" and "tvmonitor". Interestingly, we also found that such complementary merits also are effective in correcting predictions that are originally predicted incorrectly by both TaI-DPT Guo et al. (2022) and T2I, *e.g.,* in image **(c)**, both TaI-DPT Guo et al. (2022) and T2I produce the **wrong** prediction in class "person", while T2I-PAL yields the correct prediction. These performances confirm that our method, which absorbs the complementary merits of both text captions and synthesized images, is an effective way for MLR.

## H VISUALIZATION OF THE SYNTHETIC IMAGES.

Here, we visualize some synthetic images, their corresponding original images, and text descriptions in Fig. 7, where the synthetic images are generated by Stable Diffusion from the text descriptions. One can observe from the figure that the synthetic image generated from the text description preserves the key semantic information of the original image. In particular, the synthetic images are photo-realistic and diverse, providing an effective way for MLR while using only text captions. In contrast,

Table 13: **Comparison** with different visual encoders, *i.e.,* ResNet50 and ResNet101, where ↑ and ↓ indicates **improvements** and **decrements** compared with TaI-DPT Guo et al. (2022).

| | Method | MS-COCO | VOC 2007 | NUS-WIDE |
|---|---|---|---|---|
| RN50 | TaI-DPT[CVPR23] Guo et al. (2022) | 88.30 | 65.10 | 46.50 |
| | T2I-PAL(Ours) | **88.80**(0.5) ↑ | **66.10**(1.0) ↑ | **45.50**(1.0) ↓ |
| RN101 | TaI-DPT[CVPR23] Guo et al. (2022) | 88.30 | 65.40 | 45.30 |
| | T2I-PAL(Ours) | **91.50**(3.2) ↑ | **71.40**(6.0) ↑ | **47.40**(2.1) ↑ |

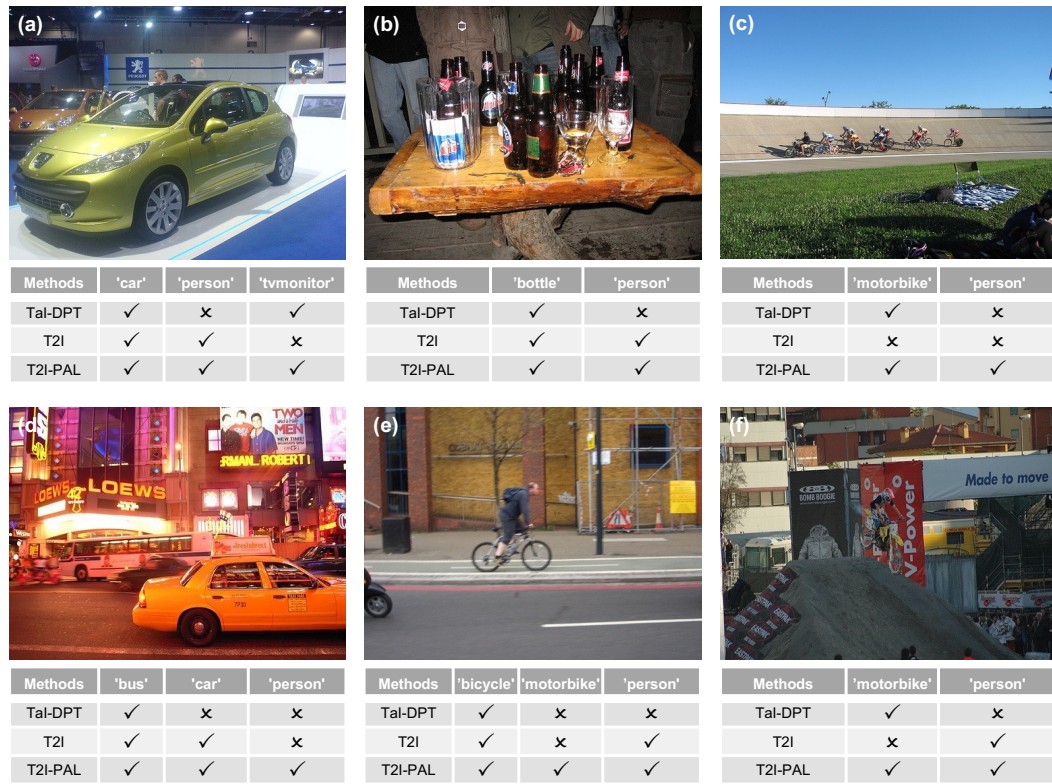

Figure 6: **Exemplars** of the complementarity of TaI-DPT Guo et al. (2022) and T2I, where ✓ and ✗ refer to the **correct** and **wrong** predictions of each class, respectively.

TaI-DPT Guo et al. (2022) that directly uses the text descriptions for parameter-efficient fine-tuning, leading to a modality gap in different feature spaces.

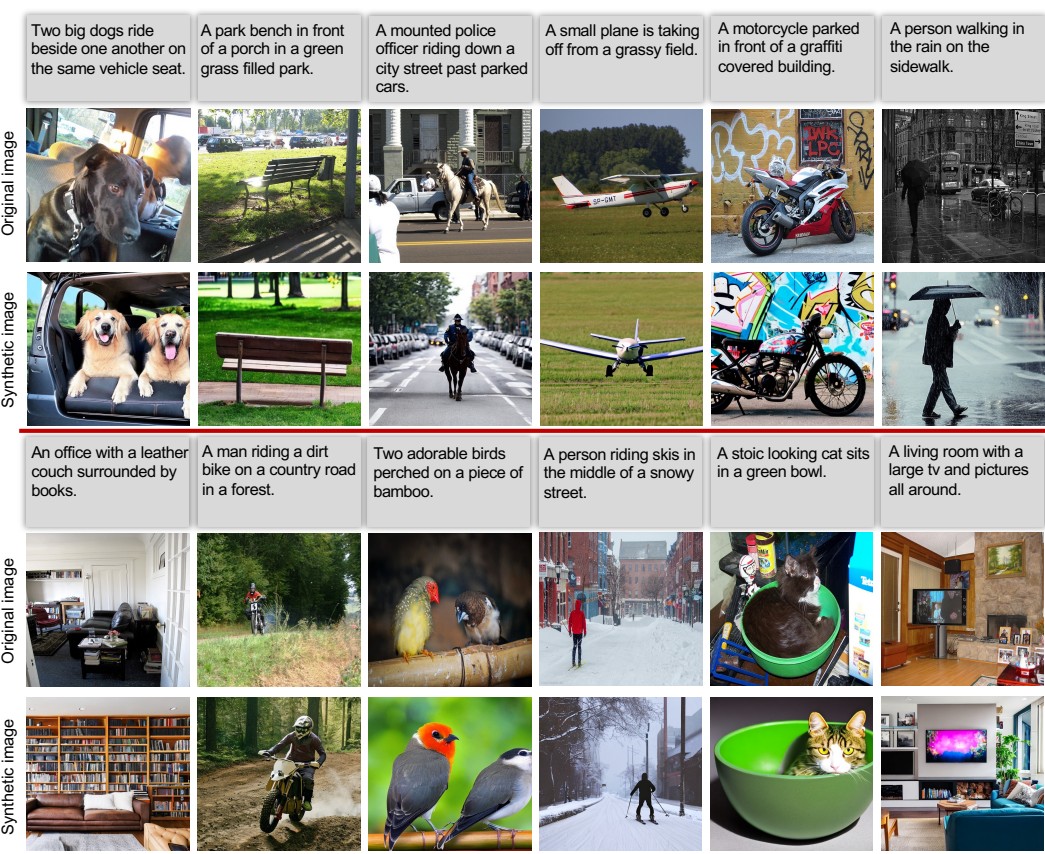

Figure 7: **Visualization** of the **synthetic** image with their corresponding **text description** and **original** image.