# OpenReview forum: "Text to Image for Multi-Label Image Recognition with Joint Prompt-Adapter Learning"
_ICLR.cc/2024/Conference — ICLR 2024 Conference Withdrawn Submission_

### Official Review · Reviewer_oaxK · 2023-10-31

**Soundness:** 3 good
**Presentation:** 1 poor
**Contribution:** 2 fair
**Rating:** 5
**Confidence:** 3

**Summary:**

The paper describes T2I-PAL, a method of leveraging pretrained text and image encoders such as CLIP, along with text-to-image generators to reduce the modality gap between text and images. The method is demonstrated on zero shot multi-label image recognition.

The crux of the idea appears to be the following:
a) mine for image captions, identify the nouns, and filter based on the classes of interest
b) construct more captions based on the nouns
c) use a text-to-image generator (e.g., Stable Diffusion) to generate images from the output of b)
d) learn an adapter to generate class embeddings to be used for cosine similarity for the final multi-label image recognition result.

Experiments are performed on MS-COCO, VOC 2007, and NUS.

**Strengths:**

The paper appears to mostly be motivated by Guo 2022, "Texts as Images in Prompt Tuning for Multi-Label Image Recognition" with the innovation that a text-to-image generator could be employed to further use additional image features.

The paper presents a useful study on the role of text-to-image generation in model training and tuning for multi-label image recognition.

Experiments are performed on zero shot, few shot and partial label settings and show a modest improvement over Guo; though that paper did not use image generation for learning prompts, but rather used text descriptions containing target categories.

**Weaknesses:**

Overall, the biggest weakness is the presentation of the paper. It's unclear exactly what the steps of the method are. The paper is lacking a clear statement of its novelty. Unclear grammar usage further obscures the intent and makes the paper a difficult read.

In the experiments, there are points that are unclear (see questions below), e.g., related to where exactly the captions come from, and how many images are generated, how many captions are synthesized, etc. Sec. B of the Appendix seemed more clear in some aspects that the main paper.

One thing I'm unconvinced about is the overall novelty of training a model on synthetic images as this has a long history in computer vision. E.g.,
Real-Time Human Pose Recognition in Parts from Single Depth Images, 2011, Shotton et al. Further, diffusion models themselves can be deployed as classifiers:
Text-to-Image Diffusion Models are Zero-Shot Classifiers https://arxiv.org/pdf/2303.15233.pdf
Your Diffusion Model is Secretly a Zero-Shot Classifier https://arxiv.org/pdf/2303.16203v3.pdf. So, in a way, the steps of generating the images and then training on those images is a way to distill the information from the text-to-image model into a classifier or embedder.
As text-to-image models become more capable and comprehensive (e.g., Stable Diffusion), essentially it's not a surprise that the model itself can act as a training dataset that can be produced on demand.

In the experiments, I would have liked to see more depth in the ablations w.r.t. the text-to-image models or other off-the-shelf components.  Does the final model have a performance gap between real images and synthetic images? Are there cases where the generation is less accurate that affect the final MLR?

Overall, I feel the paper is a useful study, but it's presentation is of a quality that it should be completely re-written and resubmitted in another venue.

Grammar / Detailed comments:
p. 3: "and then search for sentences containing at least one class name ci in C, otherwise remove them directly." Remove what?
p. 4: "stable diffusion" --> "Stable Diffusion"
p. 5: "Then, class-wise heat map ..." --> "Then the classwise heatmap ..."
p. 5: "hyper-parameter" --> "hyperparameter"
p. 5: "Accordingly, the smaller the value of alpha, the more prior knowledge needs to be acquired" --> This is unclear. Perhaps it is meant to indicate that as alpha decreases, the influence of prior knowledge increases?
p. 5: "on model performance also investigate in Suppl" --> "on model performance is also investigated in the Supplementary section"
p. 6: Datasets: unclear if the captions taken are only from the training set?
p. 6: unclear how many images are generated?
Table 2 -- Instances where other methods tie against T2I-PAL, or exceed T2I-PAL are not correctly bolded.
Fig. 3 legends for both line types need to be together.
p. 7: "As can be seen from the table that w/o. Adp " -->please rephrase
Fig. 4.  needs units on the x-axis.
are hyperparameters tuned to training set?
p. 9: "gains 3.2 improvements on" --> This needs to be rewritten. Refers to a 3.2% improvement.
Supp. Table 13 c) the image does not appear to contain a "motorbike".
Throughout: Many terms and figures are bolded for no apparent reason. E.g., "MS-COCO" is often, but not always bolded in the flow of the text.

**Questions:**

a) How are the hyperparameters tuned?
b) It's unclear whether the captions used (p. 3 "Preparation of Text Captions to Synthesize Images", p.6 "Datasets") are from the training set or test set, please clarify.
c) In experimentation, it's unclear how many training images were generated per caption, and how many captions are synthetically generated, etc.
d) Please clarify the novelty and contribution of the paper.

---

### Official Review · Reviewer_M37Z · 2023-11-01

**Soundness:** 3 good
**Presentation:** 3 good
**Contribution:** 3 good
**Rating:** 6
**Confidence:** 2

**Summary:**

The paper presents a method for parameter-efficient fine-tuning (PEFT) on the task of multi-label image recognition (MLR). It addresses the modality gap issue in existing methods by a design that involves generating training images using the Stable Diffusion model from text caption. The method shows accuracy improvement on multiple testing benchmarks for MLR.

**Strengths:**

- The paper has a clear and meaningful motivation (the modality gap in PEFT-basd methods on MLR). The proposed method directly address the issue and shows good efficacy.
- The experiments are extensive and results are competitive. The method shows consistent accuracy improvement compared with previous methods. The experiments also include a good amount of ablation studies that cover multiple important aspects of the design.
- The paper is well organized and well written.

**Weaknesses:**

- The improvement over existing methods, especially TaI-DPT, is very small and whether this improvement could be attributed to other reasons, e.g. any concern on testing data leakage in Stable Diffusion's massive training data?
- The proposed design is significantly more complex than TaI-DPT, which may be undesirable especially given the small performance improvement.

**Questions:**

- What are some evidence of previous methods suffering from modality gap and how this proposed method shows improvement in this aspect? Can there be a measurement?

---

### Official Review · Reviewer_U89w · 2023-11-01

**Soundness:** 3 good
**Presentation:** 3 good
**Contribution:** 2 fair
**Rating:** 6
**Confidence:** 4

**Summary:**

Tis paper proposes a method called T2I-PAL to address the modality gap problem in multi-label image recognition task.  By utilizing a pre-trained text-to-image generation model, T2I-PAL generates realistic and diverse images from textual descriptions thereby reducing the modality gap. To further enhance performance, the method combines both prompt tuning and  adapter learning. Experimental results demonstrate the superiority of T2I-PAL over existing methods in multi-label image recognition tasks.

**Strengths:**

I. This method does not require any original training images and does not suffer from performance degradation due to the modality gap caused by using only text captions.

II. It achieved good results in experiments and is superior to other prompt-adapter learning methods.

**Weaknesses:**

I. How to ensure that the text image generation model generates high-quality synthesized data?

II. Categories that are not in the vocabulary seem to have not been generated, and there are domain gaps between the synthesized and real images.

**Questions:**

I. The details of the experimental section were not explained clearly, such as the parts about Table 5 and Figure 4. Especially, the value in Figure 4 is not optimal for 6.

II. The author needs to confirm whether the text to image generation method uses MS-COCO and Pascal VOC datasets for training. If so, using the corresponding synthetic images is not essentially a zero-shot training configuration.

---

### Official Review · Reviewer_GRDf · 2023-11-02

**Soundness:** 2 fair
**Presentation:** 2 fair
**Contribution:** 2 fair
**Rating:** 3
**Confidence:** 4

**Summary:**

This paper presents a method, called T2I-PAL, to tackle the modality gap issue when training multi-label image recognition (MLR) models. Specifically, T2I-PAL leverages pre-trained text-to-image generation models to generate photo-realistic and diverse images from text captions. For better PEFT, the authors further combine both prompt tuning and adapter learning for enhancing classification performance.

**Strengths:**

The main idea and technical detailed are clearly presented.

**Weaknesses:**

The originality and technical contribution of this work is quite limited. Using synthetic data to enhance classification performance is not a new idea.  Prompt tuning and adapter learning have been proposed or utilized in previous works (e.g., (Guo et al., 2022) and (Zhang et al., 2022)). The authors should give more in-depth analyses or insights.

**Questions:**

The authors should explain and verify the originality and technical contribution of the proposed method.